# Genotype–environment interactions determine microbiota plasticity in the sea anemone *Nematostella vectensis*

Laura Baldassarre[1,2], Adam M. Reitzel[3], Sebastian Fraune [1]*

**1** Institut für Zoologie und Organismische Interaktionen, Heinrich-Heine Universität Düsseldorf, Düsseldorf, Germany, **2** Istituto Nazionale di Oceanografia e di Geofisica Sperimentale—OGS, Sezione di Oceanografia, Trieste, Italy, **3** Department of Biological Sciences, University of North Carolina at Charlotte, Charlotte, North Carolina, United States of America

\* fraune@hhu.de

**Data Availability Statement:** The raw sequencing data are deposited at the Sequence Read Archive (SRA) and available under the project

## Abstract

Most multicellular organisms harbor microbial colonizers that provide various benefits to their hosts. Although these microbial communities may be host species- or even genotype-specific, the associated bacterial communities can respond plastically to environmental changes. In this study, we estimated the relative contribution of environment and host genotype to bacterial community composition in *Nematostella vectensis*, an estuarine cnidarian. We sampled *N. vectensis* polyps from 5 different populations along a north–south gradient on the Atlantic coast of the United States and Canada. In addition, we sampled 3 populations at 3 different times of the year. While half of the polyps were immediately analyzed for their bacterial composition by 16S rRNA gene sequencing, the remaining polyps were cultured under laboratory conditions for 1 month. Bacterial community comparison analyses revealed that laboratory maintenance reduced bacterial diversity by 4-fold, but maintained a population-specific bacterial colonization. Interestingly, the differences between bacterial communities correlated strongly with seasonal variations, especially with ambient water temperature. To decipher the contribution of both ambient temperature and host genotype to bacterial colonization, we generated 12 clonal lines from 6 different populations in order to maintain each genotype at 3 different temperatures for 3 months. The bacterial community composition of the same *N. vectensis* clone differed greatly between the 3 different temperatures, highlighting the contribution of ambient temperature to bacterial community composition. To a lesser extent, bacterial community composition varied between different genotypes under identical conditions, indicating the influence of host genotype. In addition, we identified a significant genotype x environment interaction determining microbiota plasticity in *N. vectensis*. From our results we can conclude that *N. vectensis*-associated bacterial communities respond plastically to changes in ambient temperature, with the association of different bacterial taxa depending in part on the host genotype. Future research will reveal how this genotype-specific microbiota plasticity affects the ability to cope with changing environmental conditions.

PRJNA757926 (https://www.ncbi.nlm.nih.gov/sra/?term=PRJNA757926). Underlying raw data can be found in S1 Data.

**Funding:** This work was supported by the Human Frontier Science Program (https://www.hfsp.org/) (Young Investigators' Grant RGY0079/2016) to AR and SF and the Deutsche Forschungsgemeinschaft (https://www.dfg.de/), CRC grant 1182 "Origin and Function of Metaorganisms" (Project B1) to SF. The funders had no role in study design, data collection and analysis, decision to publish, or preparation of the manuscript.

**Competing interests:** The authors have declared that no competing interests exist.

## Introduction

Most multicellular organisms live in association with microbial symbionts [1,2]. It has been widely demonstrated that these symbionts provide various benefits for the survival and persistence of their hosts [3–5]. The quality and quantity of associated microbial species is characteristic for host species [6–8], genotype [9,10], biogeography [11–13], life stage [14–17], diet [18–20], and environmental conditions [12,13,21,22]. Starting from these evidences, many studies demonstrated that the host plays an active role in shaping its symbiont microbiota [7,23–26]. In addition to the effects of the host and the environment, the interaction between these 2 factors is also discussed as a potential factor influencing the plasticity of the microbiota [27].

*Nematostella vectensis* is a small, burrowing estuarine sea anemone found in tidally restricted salt marsh pools. The distribution of this species extends over the Atlantic and Pacific coasts of North America and the southeast coast of England [28] and its range encompasses large latitudinal variation in temperature and salinity [29]. *N. vectensis*' wide environmental tolerance and broad geographic distribution [28,30], combined with the availability of a genome sequence [31] make it an exceptional organism for exploring adaptations to variable environments. *N. vectensis* has separated sexes and it is able to reproduce both sexually through external fertilization [30,32,33] and asexually through transverse fission [28]. Although a free-swimming larval stage is present, this species is considered to have overall pretty limited dispersal abilities [34]. Seasonal population fluctuations in density may lead to frequent bottlenecks, and when gene flow between subpopulations is restricted by physical barriers, such fluctuations could result in conspicuous genetic structuring between locations over short geographic distances [34,35]. Completely or largely clonal populations exist all through the distribution range of *N. vectensis* [28,34,36]; however, microsatellite and SNP markers indicated an extensive intraspecific genetic diversity and genetic structuring between populations in their native range along the Atlantic coast of North America [37,38].

Within a single estuary, *N. vectensis* occupies tidal streams that flush with each tide or, isolated still-water high-marsh pools, that can differ substantially in a set of ecological variables including temperature and salinity [39,40]. Previous works showed that different *N. vectensis* genotypes from same natural pools within a single estuary have significantly different tolerances to oxidative stress [39] and that individuals from different field populations respond differently to same thermal conditions during lab culturing [41].

An initial categorization of the *N. vectensis* microbiota has shown that individuals from different field pools of the North American Atlantic coast have significantly different microbiota and that these differences follow a north–south gradient [12]. The different ecological conditions that distinguish these pools from each other and the genetic structuring of *N. vectensis* populations led us to hypothesize that the microbiota is a subject to local selection. In particular, locally adapted host genotypes may associate with symbionts that provide advantages at the specific ecological conditions of each native pool. Recently, we have shown that genetically identical animals differentiate their microbiota composition in response to a change in environmental temperature. By transplanting the adapted microbiota onto non-adapted animals, we demonstrated that the observed microbiota plasticity leads to increased tolerance of the animals to thermal stress [21]. However, the influence of the host genotype on the plasticity of the microbiota and thus on the ability to cope with changing environmental conditions remained unclear.

In this study, we analyzed the microbiota composition of polyps from different populations directly after sampling and after 1 month of laboratory maintenance. We first investigated which factors among ambient temperature, salinity, season, and geographic location, contribute to microbiota differentiation. The results of these analyses show that the composition of

the microbiota changes with both season and geographic location, and that these differences persist under laboratory conditions. Consistent with previous laboratory observations [12], our field data confirmed that temperature, over salinity, is correlating the most with differences in bacterial community compositions. Starting from these evidences, we investigated the influence of ambient temperature on the microbiota plasticity of 12 individual genotypes derived from 6 different populations. We found that after 3 months of laboratory culture, temperature was the factor most driving microbiota differentiation, although differences according to genotype were also detectable. In addition, we demonstrated that microbiota plasticity in relation to temperature is genotype-specific, suggesting that microbiota plasticity is also influenced by interactions between genotype and temperature.

With this study, we have taken an important step toward understanding the contribution of both local environmental conditions and host genotype in shaping the microbiota. Furthermore, we have shown that although microbial community dynamics are plastic, each genotype is associated with a microbiota that exhibits genotype-specific flexibility. These results suggest that local populations of the same species may have different abilities to adapt to environmental changes through microbiota-mediated plasticity.

## Materials and methods

### Animal sampling and culture

All experiments were carried out with polyps of *N. vectensis* (Stephenson 1935). Adult animals were collected from field populations of Nova Scotia (10/03/2016), Maine (11/03/2016, 02/06/2016, 11/09/2016), New Hampshire (11/03/2016, 02/06/2016, 11/09/2016), Massachusetts (12/03/2016, 03/06/2016, 13/09/2016), Maryland (long-term lab culture), and North Carolina (16/03/2016) by sieving them from loose sediments. Environmental parameters (air temperature, water temperature, and salinity) were also recorded at the moment of sampling and used as metadata for further analysis (see **S1 Table** for details). Half of the animals from March sampling were kept for 1 month in the laboratory, under constant, artificial conditions, at 20°C, without substrate or light, in *N. vectensis* Medium (NM), which was adjusted to 16 ppt salinity with Red Sea Salt and Millipore $H_2O$ (according to [30]). Polyps were fed 2 times a week with first instar nauplius larvae of *Artemia salina* as prey (Ocean Nutrition Micro *Artemia* Cysts 430 to 500 gr, Coralsands, Wiesbaden, Germany) and washed once a week with media pre-incubated at 20°C.

### Animal acclimation

Independently from the sampling effort described above, individually sampled polyps from 6 wild populations were asexually propagated for more than 1 year under laboratory conditions. After that, 2 strains from each original population were selected for the following experiment. Three polyps (3 replicates) for each of the 12 strains selected were placed separately into 6-well plates and let acclimate for 3 months at each of the 3 different acclimation temperatures (15, 20, and 25°C). After 3 months, the polyps were collected, frozen in liquid N, and stored at −80°C before DNA extraction and 16S sequencing.

### DNA extraction

The specimens from the field were preserved in RNAlater until DNA extraction. For the samples from the field and after 1 month of lab culture and for negative controls, gDNA was extracted with the AllPrep DNA/RNA Mini Kit (Qiagen), as described in the manufacturer's protocol. The animals from the experiment were washed 2 times with 2 ml autoclaved MQ,

instantly frozen in liquid N without liquid and stored at −80˚C until extraction. The gDNA was extracted from whole animals plus a negative control with the DNeasy Blood & Tissue Kit (Qiagen, Hilden, Germany), as described in the manufacturer's protocol. Elution was done in 50 μl and the eluate was stored at −80˚C until sequencing. DNA concentration was measured by gel electrophoresis (5 μl sample on 1.2% agarose) and by spectrophotometry through Nano-drop 3300 (Thermo Fisher Scientific).

## 16S rRNA sequencing

For each sample, the hypervariable regions V1 and V2 of bacterial 16S rRNA genes were amplified. The forward primer (5′-**AATGATACGGCGACCACCGAGATCTACAC** XXXXXXXX TATGGTAATTGT AGAGTTTGATCCTGGCTCAG-3′) and reverse primer (5′-**CAAGCAGAAGACGGCATACGAGAT** XXXXXXXX AGTCAGTCAGCC TGCTGCCTCCCGTAGGAGT-3′) contained the Illumina Adaptor (in bold) p5 (forward) and p7 (reverse). Both primers contain a unique 8 base index (index; designated as XXXXXXXX) to tag each PCR product. For the PCR, 100 ng of template DNA (measured with Qubit) were added to 25 μl PCR reactions, which were performed using Phusion Hot Start II DNA Polymerase (Finnzymes, Espoo, Finland). All dilutions were carried out using certified DNA-free PCR water (JT Baker). PCRs were conducted with the following cycling conditions (98˚C—30 s, 30 × [98˚C—9 s, 55˚C—60 s, 72˚C—90 s], 72˚C—10 min) and checked on a 1.5% agarose gel. The concentration of the amplicons was estimated using a Gel Doc XR+ System coupled with Image Lab Software (BioRad, Hercules, California, United States of America) with 3 μl of O'GeneRuler 100 bp Plus DNA Ladder (Thermo Fisher Scientific, Waltham, Massachusetts, USA) as the internal standard for band intensity measurement. The samples of individual gels were pooled into approximately equimolar subpools as indicated by band intensity and measured with the Qubit dsDNA BR Assay Kit (Life Technologies GmbH, Darmstadt, Germany). Subpools were mixed in an equimolar fashion and stored at −20˚C until sequencing. Sequencing was performed on the Illumina MiSeq platform with v3 chemistry [42]. The raw data are deposited at the Sequence Read Archive (SRA) and available under the project PRJNA757926.

## Analyses of bacterial communities

The 16S rRNA gene amplicon sequence analysis was conducted through the Qiime2 2022.8 package [43]. Adapters trimming and sequences quality filtering was performed through Dada2 [44]. Sequences with at least 100% identity were grouped into Amplicon sequence variants (ASV) and clustered against the Silva 138 reference sequence database. Samples with less than 5,000 sequences were also removed from the dataset, being considered as outliers. For the successive analysis, the number of ASVs per sample was normalized to the lowest number of reads after filtering.

Alpha-diversity represents the total number of different ASVs observed in each sample. Beta-diversity matrices were generated through Qiime2 according with the different beta-diversity metrics available (Bray–Curtis, Jaccard, Weighted-Unifrac and Unweighted-Unifrac). Statistical values of clustering were calculated using the nonparametric comparing categories methods PERMANOVA and Anosim. A Mantel test was applied to infer correlation between the different beta-diversity and environmental parameters distance matrices. The multifactorial PERMANOVA was performed through Primer 7.0.21 (https://www.primer-e.com), by testing the impact of temperature and genotype on the microbiota beta-diversity as fixed factors, since all categories of our experiment were included in the test. In order to test the different impacts between pairs of genotypes originated from the same geographic location, the genotype was nested within the location applied as random factor.

Statistical tests were performed through JASP v0.16.4 (https://jasp-stats.org). Data were subjected to descriptive analysis, and normality and variance homogeneity tests as described herein. For univariate analyses, statistical differences were tested through nonparametric Mann–Whitney U-test; for multivariate analyses, statistical significance was tested through nonparametric Kruskal–Wallis test followed by Dunn's post hoc comparisons.

Bacterial ASVs specifically associated with each genotype and each temperature was identified through LEfSe (http://huttenhower.sph.harvard.edu/galaxy) [45]. LEfSe uses the nonparametric factorial Kruskal–Wallis sum-rank test to detect features with significant differential abundance, with respect to the biological conditions of interest; subsequently LEfSe uses linear discriminant analysis (LDA) to estimate the effect size of each differentially abundant feature. Assuming that different genotypes from the same location may naturally share a number of symbionts, we only performed pairwise comparisons between genotypes from different locations. In addition to that, presence–absence calculations were performed directly on the ASV tables in order to detect bacterial ASVs that are unique for a specific genotype or AT.

## Results

### Laboratory maintenance results in loss of bacterial diversity associated with *N. vectensis* polyps

Genomic DNA samples from 168 *N. vectensis* polyps were submitted for 16S rRNA gene sequencing. While 53 samples were collected from 5 different populations (Nova Scotia, Maine, New Hampshire, Massachusetts, and North Carolina) in March 2016, the sampling in Maine, New Hampshire, and Massachusetts was repeated also in June and September (31 and 34 samples, respectively). In addition, we maintained 50 polyps sampled in March, for 1 month under laboratory conditions before we extracted gDNA. Sequencing was successful for 156 samples. A total of 25.737 different ASVs were detected, with 5.208 to 106.793 reads per sample.

Maintaining *N. vectensis* polyps for 1 month under laboratory conditions resulted in a major shift in the associated bacterial communities compared to the bacterial communities of polyps directly sampled from the field (Fig 1A and Table 1). The bacterial variability between

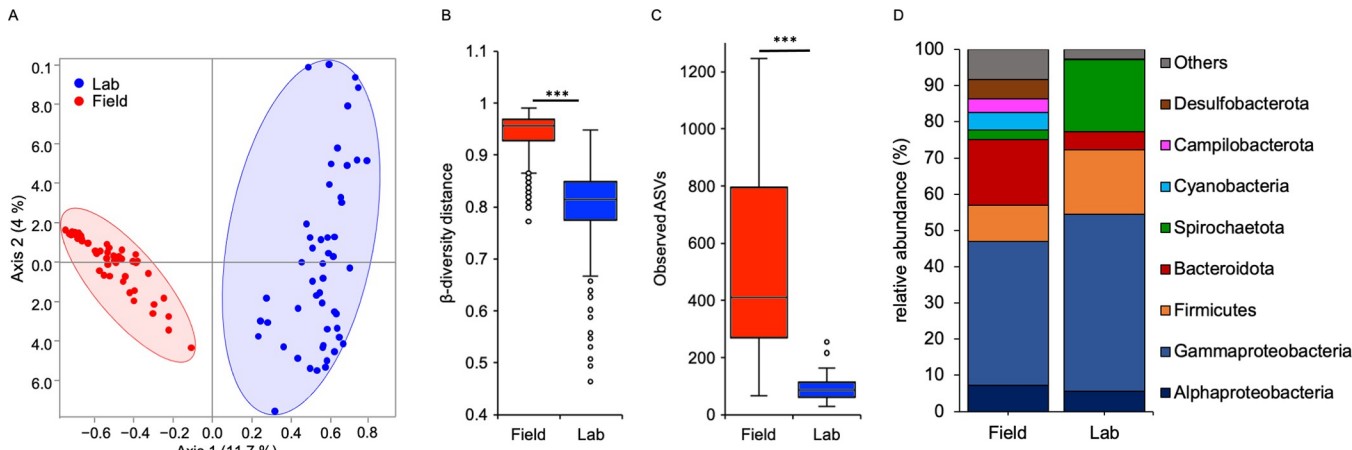

**Fig 1. Laboratory maintenance reduced bacterial diversity associated with *N. vectensis* polyps. (A)** PCoA (based on Jaccard metric, sampling depth = 5.000) illustrating similarity of bacterial communities based on sample source; **(B)** beta-diversity distance box plots of the field and lab samples; **(C)** alpha-diversity comparisons between field and lab samples (max rarefaction depth = 5.000, num. steps = 10). Differences in B and C were tested through Mann–Whitney U-test (*** = $p \leq 0.001$); **(D)** relative abundance of main bacterial groups among the 2 different samples sources. Underlying data can be found in **S1 Data**.

**Table 1. Statistical analysis determining the influence of animal laboratory maintenance on bacterial colonization.**

| | | PERMANOVA | | Anosim | |
|---|---|---|---|---|---|
| | Beta-diversity metric | pseudo-F | p-value | R | p-value |
| Source | Bray–Curtis | 13.129 | 0.001 | 0.488 | 0.001 |
| | Jaccard | 12.580 | 0.001 | 0.803 | 0.001 |
| | Weighted-Unifrac | 28.804 | 0.001 | 0.584 | 0.001 |
| | Unweighted-Unifrac | 26.693 | 0.001 | 0.950 | 0.001 |

Statistical analyses were performed (methods PERMANOVA and ANOSIM, number of permutations = 999) on each of the pairwise comparison distance matrices generated.

polyps significantly decreases during 1 month of laboratory culturing (**Figs 1B and S1**) and the alpha-diversity decreases to around 1 quarter of that observed in field sampled *N. vectensis* polyps (**Fig 1C**).

The loss of bacterial diversity in laboratory-maintained polyps became also evident by comparing the major bacterial groups (**Fig 1D**). While Cyanobacteria, Campilobacteria, and Desulfobacteria disappeared and Bacteroidota decreased in relative abundance in laboratory-maintained animals, Gammaproteobacteria, Firmicutes, and Spirochaetota increased in relative abundance (**Fig 1D**).

To determine whether bacterial communities from polyps collected from different locations reveal a biogeographic signal, and to test whether this signal is preserved in polyps maintained in the laboratory, we analyzed the 2 data sets, field and laboratory samples, separately.

## Microbial diversity in the field correlates with host biogeography and environmental factors

Analyzing the bacterial communities associated with *N. vectensis* polyps sampled in the field in March 2016, principal coordinates analysis (PCoA) revealed a clear clustering of the associated bacterial community by provenance location (**Fig 2A and 2B and Table 2**). Based on the different beta-diversity measures, geographic location explained between 56% and 83% of the bacterial variability (**Table 2**). The beta-diversity distance between samples within the same location was significantly lower than that between the different locations, stressing the clustering of the samples sharing the same provenance (**Fig 2C**).

We next investigated the influence of geographic distance, water temperature, and water salinity on a continuous scale by applying Mantel tests to each of the 5 measures of beta-diversity (**Table 3**). Mantel tests revealed that the geographic distance is the main factor impacting beta-diversity, explaining approximately 25% to 73% of the variation (**Table 3**). While both environmental factors, temperature, and salinity also correlated significantly with bacterial diversity, water temperature explained the highest proportion (**Table 3**).

In addition, alpha-diversity showed also a biogeographic signal. Polyps from the extreme northern and southern locations (Nova Scotia and North Carolina) had lower bacterial alpha-diversity than polyps from central locations (**Fig 2D**). By looking at the principal bacterial groups in the field samples, a north–south pattern was evident regarding the Gammaproteobacteria that increased in relative abundance moving from Maine through North Carolina, while Firmicutes and Desulfobacteria decreased in abundance moving in the same direction. The samples from Nova Scotia showed a different trend, with the Gammaproteobacteria and Firmicutes reaching the highest overall abundances while all the other groups the lowest (**Fig 2E**).

For the locations in which the samplings have been repeated at 3 different seasonal time points (Maine, New Hampshire, and Massachusetts), we investigated the differences in the

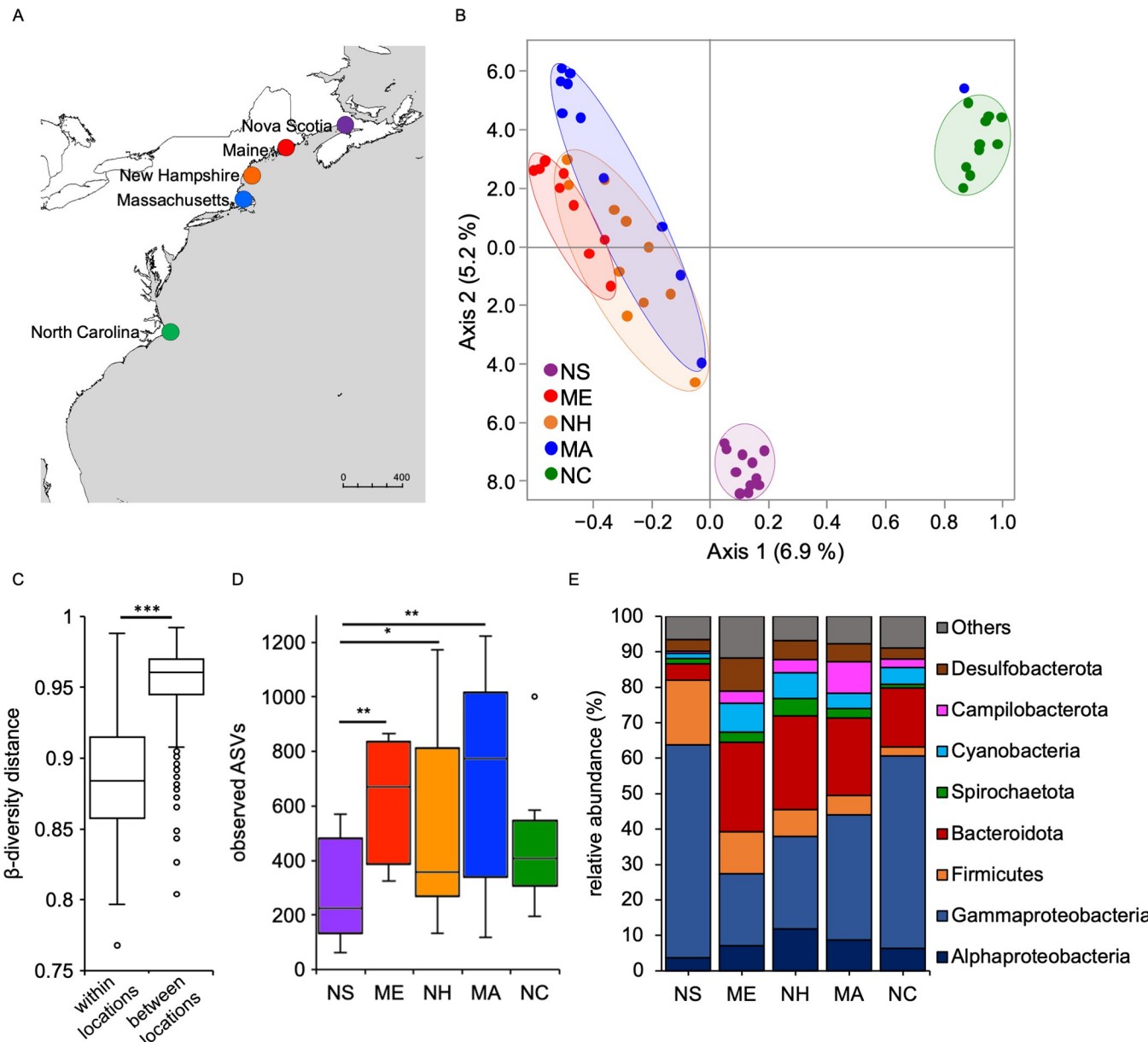

**Fig 2. Natural _N. vectensis_ populations are associated with specific microbiota.** (A) Sampling sites map. The base layer was obtained at https://www.diva-gis.org/Data. (B) PCoA (based on Jaccard metric, sampling depth = 5,000) illustrating similarity of bacterial communities based on geographic location of the March-field samples; (C) beta-diversity distance box plots within and between geographic locations, differences were tested through Mann–Whitney U-test ($*** = p \leq 0.001$); (D) alpha-diversity comparisons between geographic locations (max rarefaction depth = 5,000, num. steps = 10), differences were tested through Kruskal–Wallis test followed by Dunn's post hoc comparisons (H = 12.63, $* = p \leq 0.05$, $** = p \leq 0.01$); (E) relative abundance of main bacterial groups among different geographic locations. NS (Nova Scotia), ME (Maine), NH (New Hampshire), MA (Massachusetts), NC (North Carolina). Underlying data can be found in S1 Data.

microbiota composition according to sampling month (March, June, and September). A clustering of the samples with sampling time point was significant (**Fig 3A**), contributing up to 40% of the total difference (**Table 4**). Interestingly, the samples from June clustered in between those from March and September (**Fig 3A**), and showed a, although not significant, higher alpha-diversity than the other 2 sampling time points, suggesting a gradual shift of associated bacteria along seasons (**Fig 3B**). The Firmicutes increased in abundance moving from March

**Table 2. Statistical analysis determining the influence of geographic location on bacterial colonization in March-field samples.**

| | | PERMANOVA | | Anosim | |
|---|---|---|---|---|---|
| | Beta-diversity metric | *pseudo-F* | *p-value* | *R* | *p-value* |
| Geographic location | Bray–Curtis | 6.549 | 0.001 | 0.800 | 0.001 |
| | Jaccard | 2.684 | 0.001 | 0.833 | 0.001 |
| | Weighted-Unifrac | 7.766 | 0.001 | 0.559 | 0.001 |
| | Unweighted-Unifrac | 2.831 | 0.001 | 0.583 | 0.001 |

Statistical analyses were performed (methods PERMANOVA and ANOSIM and) on each of the pairwise comparison distance matrices generated (Number of permutations = 999).

to September in all the 3 locations (Maine, New Hampshire, and Massachusetts). Overall, the Gammaproteobacteria and Bacteroidota were more abundant in March samples, while Spirochaetota and Cyanobacteria were more abundant and Gammaproteobacteria less abundant in the samples from June, respectively (**Fig 3C**).

## *N. vectensis* polyps cultured in the laboratory maintain population-specific microbiota

To test whether the biogeographic signal of the bacterial communities associated with polyps is maintained under laboratory conditions, we analyzed the laboratory samples separately (**Fig 4**). A clear clustering of the samples according with the provenance location was still present and become even more evident after 1 month under laboratory conditions (**Fig 4A and 4B**). All the ANOVA comparisons performed and the Mantel tests were highly significant ($p < 0.001$) (**Table 5**), and showed that the provenance geographic location explained between 55% and 74% of the beta-diversity difference for the lab samples, proving that the population-specific bacterial fingerprints were maintained (**Table 5**). The beta-diversity distance between samples originating from the same location was significantly lower than that between the different locations, stressing the clustering of the samples sharing the same provenance (**Fig 4B**). For the lab samples, the alpha-diversity was also the highest in the samples from the

**Table 3. Statistical analysis determining the influence of geographic distance, field temperature, and salinity on bacterial colonization.**

| | | Mantel test | |
|---|---|---|---|
| Parameter | Beta-diversity metric | Mantel *r* | Mantel *P* |
| Geographic distance | Bray–Curtis | 0.594 | 0.001 |
| | Jaccard | 0.732 | 0.001 |
| | Weighted-Unifrac | 0.253 | 0.001 |
| | Unweighted-Unifrac | 0.400 | 0.001 |
| Temperature | Bray–Curtis | 0.568 | 0.001 |
| | Jaccard | 0.630 | 0.001 |
| | Weighted-Unifrac | 0.289 | 0.001 |
| | Unweighted-Unifrac | 0.400 | 0.001 |
| Salinity | Bray–Curtis | 0.235 | 0.001 |
| | Jaccard | 0.197 | 0.001 |
| | Weighted-Unifrac | 0.258 | 0.001 |
| | Unweighted-Unifrac | 0.155 | 0.006 |

Mantel tests were performed between the 3 different parameters distance matrices and the beta-diversity matrices generated. (Number of permutations = 999)

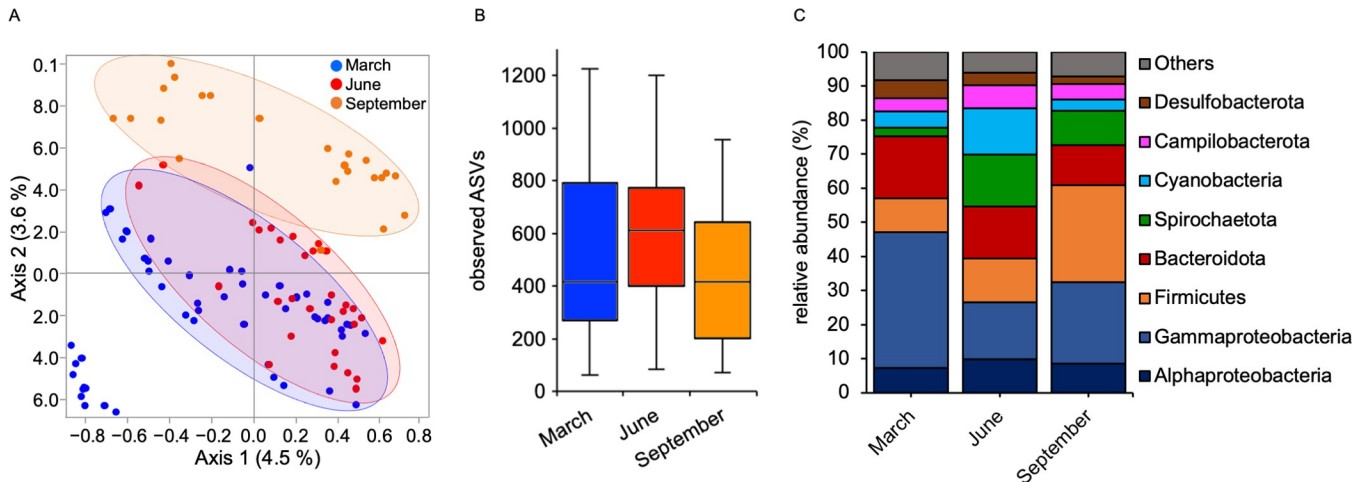

**Fig 3. Natural microbiota in *N. vectensis* vary according to season.** (A) PCoA (based on Jaccard metric, sampling depth = 5,000) illustrating similarity of bacterial communities based on sampling month; (B) alpha-diversity comparisons between sampling months (max rarefaction depth = 5,000, num. steps = 10), differences were tested through Kruskal–Wallis test (not significant); (C) relative abundance of main bacterial groups among different sampling months. Underlying data can be found in S1 Data.

intermediate locations (**Fig 4C**). Animals from the extreme locations (Nova Scotia and North Carolina) where colonized by the highest abundances of Firmicutes and Gammaproteobacteria, respectively, while those from the central latitudes were associated mainly with greater abundances of Bacteroidota and Spirochaetota (**Fig 4D**).

## Under different temperatures, *N. vectensis* maintains genotype-specific microbiota

The variation of bacterial communities associated with *N. vectensis* polyps in the field correlated mostly with ambient water temperature (**Table 3**). Based on these findings, we aimed to measure experimentally the contribution of temperature and host genotype and their interaction on the microbiota composition. We selected in total 12 genotypes originating from 6 different geographic locations (2 genotypes/location) (**Fig 5A**). To be able to maintain each genotype at different ambient temperatures, we clonally propagated the polyps to reach at least 9 clones/genotype. Subsequently, we maintained each genotype at 3 different temperatures (15, 20, and 25˚C, *n* = 3) for 3 months (**Fig 5A**). Nine polyps out of 108 did not survive the treatment. Interestingly, culturing at high temperature (25˚C) resulted in higher mortality in animals from Nova Scotia, New Hampshire, and Massachusetts, while animals from Maine had the highest mortality at low temperatures (15 and 20˚C) (**S2 Fig**).

**Table 4. Statistical analysis determining the influence of season on bacterial colonization.**

| | | PERMANOVA | | ANOSIM | |
|---|---|---|---|---|---|
| | **Beta-diversity metric** | *pseudo-F* | *p-value* | *R* | *p-value* |
| **Season** | Bray–Curtis | 7.817 | 0.001 | 0.295 | 0.001 |
| | Jaccard | 3.030 | 0.001 | 0.228 | 0.001 |
| | Weighted-Unifrac | 11.493 | 0.001 | 0.405 | 0.001 |
| | Unweighted-Unifrac | 3.827 | 0.001 | 0.207 | 0.001 |

Statistical analyses were performed (methods PERMANOVA and ANOSIM, number of permutations = 999) on each of the pairwise comparison distance matrices generated.

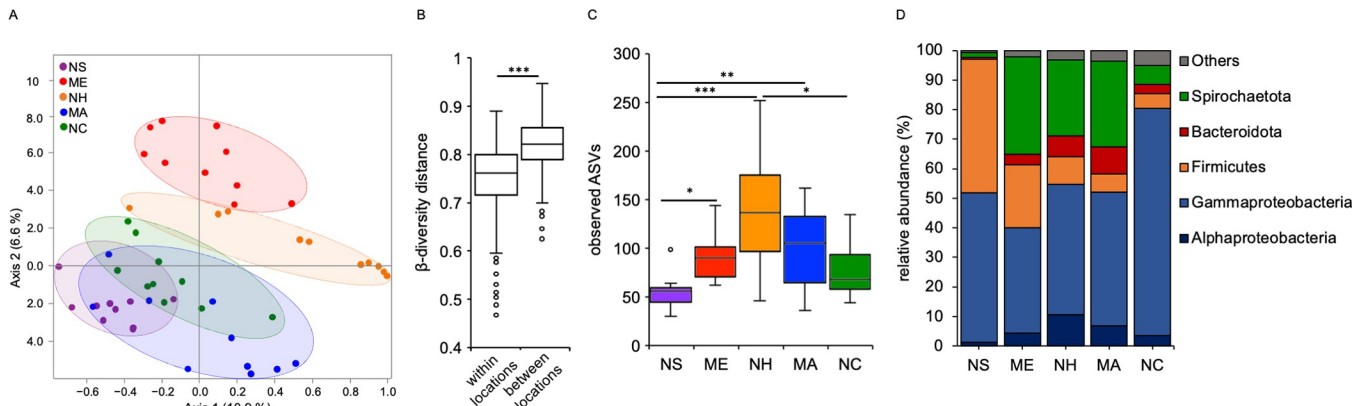

**Fig 4. Population-specific microbiota are maintained under laboratory conditions.** (A) PCoA (based on Jaccard metric, sampling depth = 5,000) illustrating similarity of bacterial communities based on geographic population; (B) beta-diversity distance box plots of the lab samples within and between geographic locations, differences were tested through Mann–Whitney U-test (*** = $p \leq 0.001$); (C) alpha-diversity comparisons between geographic locations (max rarefaction depth = 5,000, num. steps = 10); (D) relative abundance of main bacterial groups among different geographic locations. Differences were tested through Kruskal–Wallis test followed by Dunn's post hoc comparisons (H = 18.35, * = $p \leq 0.05$, ** = $p \leq 0.01$, *** = $p \leq 0.001$). NS (Nova Scotia), ME (Maine), NH (New Hampshire), MA (Massachusetts), NC (North Carolina). Underlying data can be found in S1 Data.

After 3 months of culturing at different temperatures, gDNA from 99 polyps were submitted for 16S rRNA gene sequencing. A total of 985 different ASVs were detected, with the number of reads per sample ranging between a maximum of 65.402 and a minimum of 15.850. After setting the minimum number of reads/sample at 15.800, 92 samples remained for the successive analyses.

PCoA revealed that ambient temperature explained most of the detected bacterial diversity associated with the polyps (between 15% and 59% diversity explained) (**Fig 5B** and **Table 6**), while no significant differences in beta-diversity distances were evident between the 3 different temperatures (**S3A Fig**). While principal component 1 (PC1) mostly separates samples according to the ambient temperature (**Fig 5B**), PC2 mostly explains variations within the different genotypes (**Fig 5C**). The ANOSIM results indicated that host genotype contributed between 13% and 22% to the total bacterial diversity observed (**Table 6**). The alpha-diversity slightly increased, although not significantly, from the 15°C samples through the 25°C ones (**S3B Fig**), no clear pattern from the host genotypes on the alpha-diversity analysis was evident (**S4 Fig**).

Interestingly, comparison of the beta-diversity distances of the different genotypes (**Fig 5D**) revealed that they differ significantly (Kruskal–Wallis $p < 0.001$) in terms of microbiota flexibility (**S2 Table**). These results suggest that each genotype is endowed with a microbiota that

**Table 5. Statistical analysis determining the influence of geographic distance and geographic location on bacterial colonization in laboratory-maintained populations.**

| | | PERMANOVA | | Anosim | | Mantel test | |
|---|---|---|---|---|---|---|---|
| | Beta-diversity metric | *pseudo-F* | *p-value* | *R* | *p-value* | Mantel *r* | Mantel *P* |
| **Geographic location** | Bray–Curtis | 10.653 | 0.001 | 0.736 | 0.001 | 0.608 | 0.001 |
| | Jaccard | 2.986 | 0.001 | 0.604 | 0.001 | 0.352 | 0.001 |
| | Weighted-Unifrac | 8.902 | 0.001 | 0.551 | 0.001 | 0.433 | 0.001 |
| | Unweighted-Unifrac | 3.753 | 0.001 | 0.599 | 0.001 | 0.384 | 0.001 |

Statistical analyses were performed (methods PERMANOVA and ANOSIM) on each of the pairwise comparison distance matrices generated according with provenance geographic location. Mantel test was performed between the geographic location distance matrix and the different beta-diversity matrices. (Number of permutations = 999)

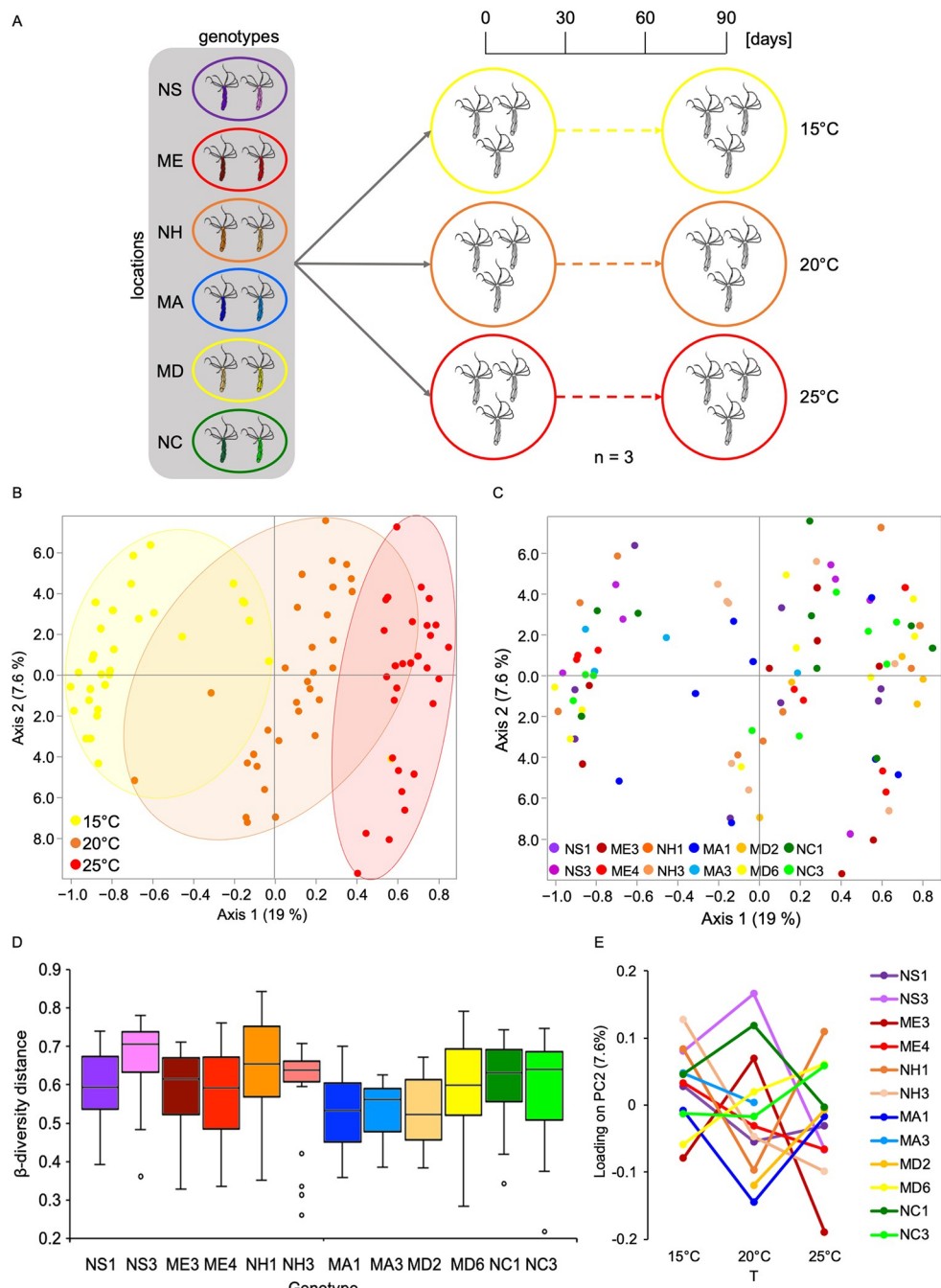

**Fig 5. Influence of host genotype and temperature on bacterial colonization.** (A) Experimental design, 2 genotypes for each geographic location were kept in 3 replicates at 3 different temperatures for 3 months; (B) PcoA (based on Jaccard metric, sampling depth = 15,800) illustrating similarity of bacterial communities based on ambient temperature; (C) PCoA (based on Jaccard metric, sampling depth = 15,800) illustrating similarity of bacterial communities based on host genotype; (D) beta-diversity distance box plots between different genotypes (Jaccard metric, sampling depth = 15,800), differences were tested through Kruskal–Wallis test (H = 38.91, $p$ = < 0.001); for clarity the Dunn's post hoc comparisons are reported in S2 Table. (E) Reaction norms plotting average principal component 2 eigenvalues for each of the 12 genotypes at each temperature. NS (Nova Scotia), ME (Maine), NH (New Hampshire), MA (Massachusetts), MD (Maryland), NC (North Carolina), numbers near the location abbreviations indicate the different genotypes. Underlying data can be found in S1 Data.

**Table 6. Statistical analysis determining the influence of host genotype and temperature on bacterial colonization in experimental animals.**

|  |  | PERMANOVA | | Anosim | |
| --- | --- | --- | --- | --- | --- |
| Parameter | Beta-diversity metric | *pseudo-F* | *p-value* | *R* | *p-value* |
| Temperature | Bray–Curtis | 12.991 | 0.001 | 0.497 | 0.001 |
|  | Jaccard | 11.027 | 0.001 | 0.591 | 0.001 |
|  | Weighted-Unifrac | 5.630 | 0.001 | 0.154 | 0.001 |
|  | Unweighted-Unifrac | 8.438 | 0.001 | 0.376 | 0.001 |
| Genotype | Bray–Curtis | 2.372 | 0.001 | 0.219 | 0.001 |
|  | Jaccard | 1.773 | 0.001 | 0.132 | 0.001 |
|  | Weighted-Unifrac | 3.041 | 0.001 | 0.225 | 0.001 |
|  | Unweighted-Unifrac | 1.869 | 0.001 | 0.155 | 0.001 |

Statistical analyses were performed (methods PERMANOVA and ANOSIM, number of permutations = 999) on each of the pairwise comparison distance matrices generated.

exhibits genotype-specific flexibility. In particular, we identified genotypes whose microbiota exhibit low flexibility (e.g., MA1 and MD2), in contrast to genotypes whose microbiota exhibit high flexibility (e.g., NS3 and NH3).

In order to detect genotype-specific bacterial adjustments to temperature variation, we performed a multifactorial PERMANOVA, by testing the influence of the genotypes within each provenance location separately. The results revealed that genotype x temperature interactions significantly influenced microbial plasticity despite the possible genotype similarities within the same location (**Table 7**). Plotting the average PC2 eigenvalues of each genotype at the 3 different ambient temperatures (**Fig 5E**) indicated that the microbial plasticity differed between the 12 different genotypes. Interestingly, the adjustments in bacterial diversity within the 12 genotypes can be divided in 2 main patterns (**S6A and S6B Fig**). Together, these results suggest different metaorganism strategies to cope with environmental changes.

In a further step, we aimed to detect indicator taxa specifically associated with ambient temperature and genotypes (**Fig 6** and **S3** and **S4 Tables**). Through LEfSe, we were able to detect indicator ASVs that are overrepresented in each sample category in comparison with all the

**Table 7. Statistical analysis determining the influence of host genotype x temperature interaction on bacterial colonization in experimental animals.**

|  |  | PERMANOVA | |
| --- | --- | --- | --- |
| Parameter | Beta-diversity metric | *pseudo-F* | *P value* |
| Temperature * Genotype | Bray–Curtis | 2.260 | 0.0001 |
|  | Jaccard | 1.823 | 0.0001 |
|  | Weighted-Unifrac | 2.342 | 0.0001 |
|  | Unweighted-Unifrac | 1.775 | 0.0001 |
| Temperature | Bray–Curtis | 10.373 | 0.0001 |
|  | Jaccard | 6.038 | 0.0001 |
|  | Weighted-Unifrac | 4.903 | 0.0002 |
|  | Unweighted-Unifrac | 5.453 | 0.0001 |
| Genotype | Bray–Curtis | 1.738 | 0.0011 |
|  | Jaccard | 1.921 | 0.0001 |
|  | Weighted-Unifrac | 1.620 | 0.0183 |
|  | Unweighted-Unifrac | 1.875 | 0.0001 |

Multifactorial PERMANOVA test was performed on each of the beta-diversity distance matrices generated. (Number of unrestricted permutations = 9,999; type I (sequential) sums of squares; temperature and genotype as fixed factors, genotype nested within location as random factor.)

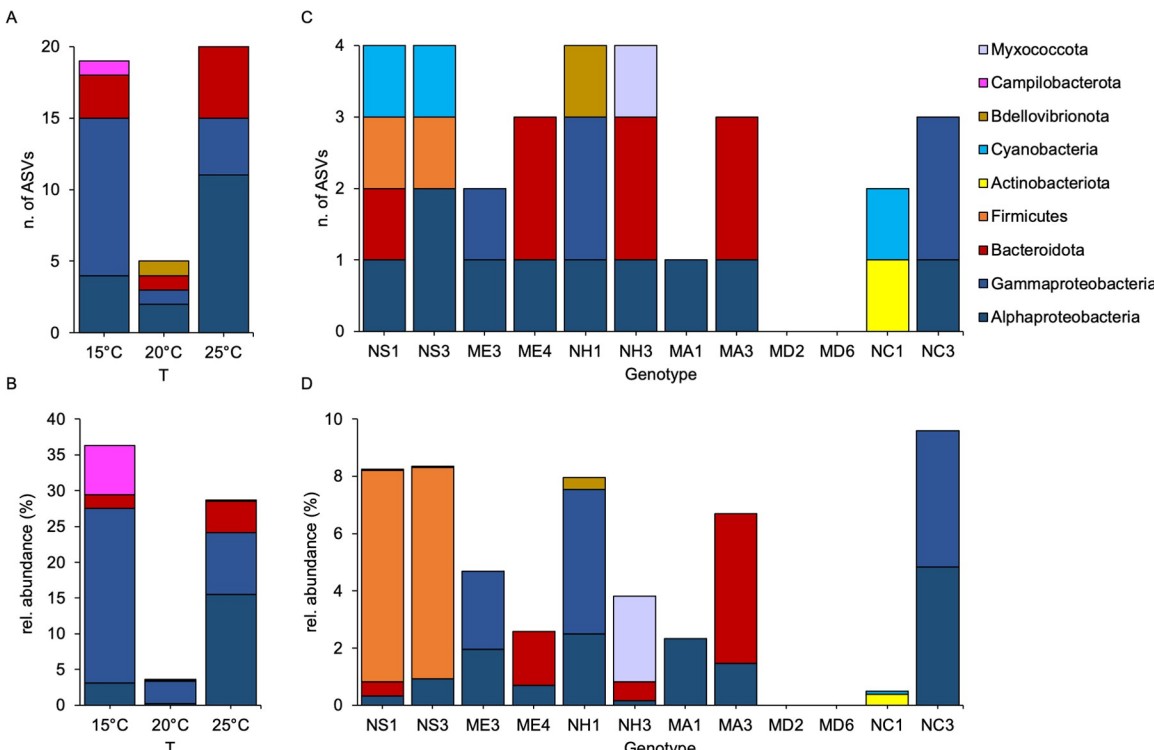

**Fig 6. Bacterial ASVs representative of host genotype and acclimation temperature.** Number of bacterial ASVs overrepresented at each temperature (A and B) and in each genotype (C and D) compared to the others, divided by major groups. Absolute ASV number (A and C), relative ASVs abundances on the total number of reads (B and D). Underlying data can be found in S1 Data.

others. We observed that extreme ambient temperatures showed higher numbers of unique associated ASVs (**Fig 6A**). Interestingly, calculating the relative abundance of indicator ASVs (**Fig 6B and 6D**) revealed that around 36% and 29% of bacterial abundance at 15˚C and 25˚C, respectively, were represented by temperature-specific ASVs. In contrast, genotype-specific ASVs represented on average 5% of the bacterial total abundance, while the 2 genotypes isolated from MD (the only long-term lab culture) did not show any genotype-specific ASV (**Fig 6D**). Interestingly, genotypes isolated from the same location show similarities in terms of specific ASVs and their relative abundances, and notably NS1 and NS3 share 3 out of their 4 genotype-specific ASVs (**Fig 6C and 6D**). These results suggest that genotypes from the same locations might be close relatives.

## Discussion

### Environmental factors can explain most but not all variability of *N. vectensis*-associated microbiota

To estimate the contributions of both environmental factors and genotype to the bacterial diversity associated with *N. vectensis*, we started with a huge sampling effort to collect individuals of *N. vectensis* from multiple populations of the US Atlantic coast along a north–south gradient of more than 1,500 km and correlated the microbial composition data to the environmental factors temperature and salinity. In addition, we sampled individuals from 3 populations also in 3 different seasons. Our results showed that temperature and salinity, although explaining a similar percentage of the observed variability, could not explain all of the observed

bacterial variation. In addition, we showed that the associated microbial community changes gradually along a temporal pattern during the year. Previous studies in corals have also shown that associated bacterial communities change depending on the season [46–49], e.g., due to changes in dissolved oxygen concentrations and rainfall [47]. In addition, seasonal changes in host physiology associated with winter quiescence may drive microbiota diversity [49]. Besides these cues, natural seasonal fluctuations in bacterial communities can also impact the availability of certain symbiotic species [50].

## Maintenance in the laboratory reduces bacterial diversity but preserves population-specific bacterial signatures

After sampling polyps from the wild, we additionally kept individuals of *N. vectensis* from each population under constant laboratory conditions for 1 month and compared these samples to those sampled directly from the field in terms of microbial diversity. In accordance with what was previously found from studies on lab mice [51], insects [52–54], and corals [55,56] laboratory-reared *N. vectensis* individuals host a significantly lower bacterial diversity than in the wild. Interestingly, the homogenous lab environment did not eliminate the original differences in bacterial colonization observed in the animals directly samples from the field. Surprisingly, the population-specific signature became even more evident in the laboratory-maintained animals. These results indicate that the bacterial diversity loss mainly affects bacteria that are not responsible for the population-specific signature. Therefore, bacteria that are lost under laboratory condition most likely are loosely associated environmental bacteria, food bacteria, or might stem from taxa that are only transiently associated with the host [57,58]. In future studies, the amount of bacterial sequences derived from dead bacteria or eDNA could be reduced by sequencing bacterial RNA instead of DNA. However, bacteria that are persisting during laboratory maintenance most likely represent bacteria that are functionally associated with *N. vectensis* [21] and might have co-evolved with its host [57,59,60].

## Genotype x environment interactions shape microbiota plasticity of *N. vectensis*

For several animal and plant species, it has been observed that associated microbial community dissimilarities increase with geographical distance [61]. Host selection, environmental filtering, microbial dispersal limitation, and microbial species interactions have all been suggested as key drivers of host-microbial composition in space and time [62]. Also a previous study in *N. vectensis* evidenced that individuals from different populations harbor distinct microbiota [12].

In order to disentangle the contribution of the host, the environment, and their interaction on the microbiota composition in *N. vectensis*, we selected 12 genotypes from 6 different field populations and kept clones of each genotype for 3 months under different temperatures. We found bacterial taxa that are associated with both specific genotypes and specific temperature conditions. These results suggest that both intrinsic and extrinsic factors shape the host-associated microbiota, although environmental conditions appear to have a stronger influence. In contrast to previous observations in corals [10,63], where host genotype had a greater impact on microbiota composition than environmental conditions, in our study, we observed that environmental conditions (in this case, temperature) had a greater effect on microbiota than genotype. Similar results were shown in fire coral clones, where both host genotype and reef habitat contributed to bacterial community variabilities [64]. Genomic function predictions suggested that environmentally determined taxa lead to functional restructuring of the microbial metabolic network, whereas bacteria determined by host genotype are functionally

redundant [64]. As previously suggested [65], these observations confirm that both environmental and host factors are drivers of associated microbial community composition and that different genotype x environment combinations can create unique microhabitats suitable for different microbial species with different functions.

One mechanism by which host selection can occur is through innate immunity, e.g., the secretion of antibiotic compounds via the mucus layer that target non-beneficial or pathogenic microbes [7,23,24,66]. Our results suggest that *N. vectensis* also plays an active role in shaping its symbiotic microbiota in response to environmental variability and that these mechanisms depend on genotypic differences and local adaptation.

## Microbial plasticity is linked to animal adaptations

Differences in prokaryotic community composition in different environments have been documented in many other marine invertebrates and are considered to reflect local acclimation [10,67–69]. We have recently shown that the restructuring of microbial communities due to temperature acclimation is an important mechanism of host plasticity and adaptation in *N. vectensis* [21]. The higher thermal tolerance of animals acclimated to high temperature could be transferred to non-acclimated animals through microbiota transplantation [21]. In our study, high temperature conditions were particularly challenging for some genotypes native to north habitats, where they experience colder climate. Whether this is the result of local adaptation of the host to colder temperatures or the symbiotic microbiota, needs to be clarified. We also observed that the bacterial species richness increases in intermediate latitudes, seasons, and temperature, while it decreases at the extremes, suggesting a dynamic and continuous remodeling of the microbiota composition along environmental conditions gradients.

Evidence from reciprocal transplantation experiments in corals followed by short-term heat stress suggests also that coral-associated bacterial communities are linked to variation in host heat tolerance [70] and that associated bacterial community structure responds to environmental change in a host species-specific manner [71]. Here, we show that not only do different species exhibit different microbial flexibility, but also genotypes can differ in the flexibility of their microbiota.

We hypothesize that host organisms may evolve faster than on their own due to plastic changes in their microbiota. Rapidly dividing microbes are predicted to undergo adaptive evolution within weeks to months. Adaptation of the microbiota can occur via changes in absolute abundances of specific members, acquisition of novel genes, mutation, and/or horizontal gene transfer [14,69,72–74]. Here, we provide evidence for genotype-specific microbial plasticity and flexibility, leading to genotype-specific restructuring of the microbial network in response to environmental stimuli. Together, these results may indicate that the genotype-specific bacterial colonization reflects local adaptation. Future studies will reveal whether lower plasticity and flexibility of the microbiota is associated with lower adaptability to changing environmental conditions and which host factors determine the plasticity and flexibility of the microbiota. In particular, genotypes adapted to highly variable environments might favor flexibility over fidelity regarding the associated microbiota composition; conversely, under more stable conditions, less dynamic and stricter association might be advantageous [75].

## Supporting information

**S1 Table. Metadata and environmental data at sampling time points.** For Nova Scotia, Maine, and New Hampshire in March, the temperatures were inferred from previous weather reports (Reitzel and colleagues, 2013).
(DOCX)

**S2 Table. Dunn's post hoc comparisons of the Kruskal–Wallis test performed on data represented in Fig 5D.** NS (Nova Scotia), ME (Maine), NH (New Hampshire), MA (Massachusetts), MD (Maryland), NC (North Carolina), numbers near the location abbreviations indicate the different genotypes; colors represent the different significance (yellow = $p \leq 0.05$, blue = $p \leq 0.01$, green = $p < 0.001$).
(DOCX)

**S3 Table. Temperature-unique ASVs.** The (+) present in all the replicates of the indicated temperature and in none of the other temperatures, (-) absent in at least 1 replicate of the indicated temperature or present in at least 1 replicate of any other temperature.
(XLSX)

**S4 Table. Genotype-unique ASVs.** The (+) present in all the replicates of the indicated genotype and in none of the other genotypes, (-) absent in at least 1 replicate of the indicated genotype or present in at least 1 replicate of any other genotype.
(XLSX)

**S1 Fig. PCoA illustrating similarity of bacterial communities based on sample source.** (Jaccard metric, sampling depth = 5,000). Underlying data can be found in S1 Data.
(DOCX)

**S2 Fig. Number of clones for each provenance location that survived at the 3 different temperatures.** Underlying data can be found in S1 Data.
(DOCX)

**S3 Fig. Beta-diversity and alpha-diversity distance comparisons between temperatures.** (A) Beta-diversity distance box plots between different temperatures (Jaccard metric, sampling depth = 15,800); (B) alpha-diversity comparisons between temperatures (max rarefaction depth = 15,800, num. steps = 10). Differences were tested through Kruskal–Wallis test (not significant). Underlying data can be found in S1 Data.
(DOCX)

**S4 Fig. Alpha-diversity comparisons between polyp genotypes.** (max rarefaction depth = 15,800, num. steps = 10) (Jaccard metric, sampling depth = 15,800), differences were tested through Kruskal–Wallis test (H = 38.91, $p = < 0.001$); for clarity the Dunn's post hoc comparisons are reported in the table. NS (Nova Scotia), ME (Maine), NH (New Hampshire), MA (Massachusetts), MD (Maryland), NC (North Carolina), numbers near the location abbreviations indicate the different genotypes. Underlying data can be found in S1 Data.
(DOCX)

**S5 Fig. Relative abundance of main bacterial groups among the different genotypes at the three temperatures.** NS (Nova Scotia), ME (Maine), NH (New Hampshire), MA (Massachusetts), MD (Maryland), NC (North Carolina), numbers near the location abbreviations indicate the different genotypes. Underlying data can be found in S1 Data.
(DOCX)

**S6 Fig. Reaction norms.** A and B show the same samples of Fig 5E divided by similar reaction norms. NS (Nova Scotia), ME (Maine), NH (New Hampshire), MA (Massachusetts), MD (Maryland), NC (North Carolina), numbers near the location abbreviations indicate the different genotypes. Underlying data can be found in S1 Data.
(DOCX)

**S1 Data. Raw data accompanying figures.**
(XLSX)

## Acknowledgments

We thank Katja Cloppenborg-Schmidt (CRC 1182 project Z3) for preparing the 16S rRNA gene library and the Institute of Clinical Molecular Biology in Kiel for providing sequencing services. We thank Jan Taubenheim for the statistical counseling and support.

## Author Contributions

**Conceptualization:** Laura Baldassarre, Adam M. Reitzel, Sebastian Fraune.

**Data curation:** Laura Baldassarre.

**Formal analysis:** Laura Baldassarre, Sebastian Fraune.

**Funding acquisition:** Adam M. Reitzel, Sebastian Fraune.

**Investigation:** Laura Baldassarre, Sebastian Fraune.

**Methodology:** Laura Baldassarre, Adam M. Reitzel.

**Project administration:** Sebastian Fraune.

**Resources:** Adam M. Reitzel, Sebastian Fraune.

**Supervision:** Sebastian Fraune.

**Validation:** Sebastian Fraune.

**Visualization:** Laura Baldassarre, Sebastian Fraune.

**Writing – original draft:** Laura Baldassarre, Sebastian Fraune.

**Writing – review & editing:** Laura Baldassarre, Adam M. Reitzel, Sebastian Fraune.

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
