## [Editor Report · Decision Letter 0]

22 Jun 2022

Dear Dr. Fraune, 

Thank you for submitting your manuscript entitled "Genotype-environment interactions determine microbiota plasticity in Nematostella vectensis" for consideration as a Research Article by PLOS Biology.

Your manuscript has now been evaluated by the PLOS Biology editorial staff, as well as by an academic editor with relevant expertise, and I am writing to let you know that we would like to send your submission out for external peer review.

Once your full submission is complete, your paper will undergo a series of checks in preparation for peer review. After your manuscript has passed the checks it will be sent out for review. To provide the metadata for your submission, please Login to Editorial Manager (https://www.editorialmanager.com/pbiology) within two working days, i.e. by Jun 24 2022 11:59PM.

Kind regards,

Paula

---

Senior Editor

PLOS Biology

---

## [Decision Letter · Decision Letter 1]

31 Aug 2022

Dear Dr. Fraune,

Thank you for your patience while your manuscript "Genotype-environment interactions determine microbiota plasticity in Nematostella vectensis" was peer-reviewed at PLOS Biology. It has now been evaluated by the PLOS Biology editors, an Academic Editor with relevant expertise, and by several independent reviewers. 

In light of the reviews, which you will find at the end of this email, we would like to invite you to revise the work to thoroughly address the reviewers' reports.

In particular, we think that technical details and study design need to be more explicitly stated within the manuscript and figures. On the technical aspects, we recommend that you do the updated analysis for ASVs instead of OTUs that could be hosted in the supplementary material as additional analysis if they corroborate the current analysis. Please also address the rest of the reviewers' issues. 

Given the extent of revision needed, we cannot make a decision about publication until we have seen the revised manuscript and your response to the reviewers' comments. Your revised manuscript is likely to be sent for further evaluation by all or a subset of the reviewers.

**IMPORTANT - SUBMITTING YOUR REVISION**

*Re-submission Checklist*

*Published Peer Review*

*PLOS Data Policy*

*Blot and Gel Data Policy*

Sincerely,

Paula

Senior Editor

PLOS Biology

REVIEWS:

Reviewer #1: Ashley M Dungan. Bacteria symbiosis and coral hosts.

Reviewer #2: Biotic forces in evolution and ecology.

Reviewer #3: Animal-microbe interactions and ecosystems.

Reviewer #1: Line specific comments: 

L132: Are you saying that all only March animals were exposed to lab conditions? What are "standard lab conditions"? Maybe site a paper or say "refer below." The sampling text needs a little more clarity or perhaps a small figure. Its an important part of the story to know what samples are from the field vs lab, and then which animals were exposed to the temperature stress.

L134: Ok - so you collected animals at a variety of temperatures (ranging -2.2 to 22.4 C according to Table S1) but then held them all in lab conditions at 20C? How did you ramp that? How did you consider this stress in your future experiments? How do you determine that 1 month was enough to be acclimated?

L135: I would prefer a psu or ppt value here instead. 

L142: How do you know they were two different strains?

L148: Not great that you used two different kits for gDNA extraction - especially when you want to compare the samples. Worthwhile to mention this possible point of difference in the discussion (or perhaps you'll find a paper that has tested the differences between these two kits).

L159: I don't see any mention of controls - seawater controls, Artemia feed, or at least extraction blanks. Especially because you used two different kits. 

L182: The data was processed with QIIME (v1)? What reference database? It's also not idea that you used OTUs. I would suggest rerunning in QIIME2 (we're currently up to v2022.2) as your data was processed in a version older than 2017 and A LOT has changed when it comes to metabarcoding in the last 5 years. While I don't object to the OTU model, amplicons should be binned at a higher resolution (i.e. 99% identity) or just use ASV. 

L194: The statistical capability of QIIME2 (which is confusing that you used this when you analyzed your data in QIIME1) is very limited. There is no capacity to add random effects among other things. The data needs to be exported and analysed in external software - R would be great. 

L325: The title of figure 4 is "Population-specific microbiota are maintained under laboratory conditions," except no part of this figure compares a given geographic region over time in laboratory conditions. 

L339: There is nothing in the methods to explain how you knew they were distinct genotypes. This needs to be explained. 

L370: This figure should be called Figure 5. I like Fig. 5A - could you have some version of that above to help explain the sampling system? Perhaps add to 5A that these were all collected in March, so the same as time point 1. Something else I don't quite understand - since these animals were collected from a range of geographic sites, did each of their "ambient" temperatures correspond to collection temperatures? I think you could get rid of Figures 5C-G. They don't add to your story and the data you present here can be sufficiently provided as text. 

L411: There are also plenty of cnidarian studies that show reductions in microbiota diversity from the field to the lab. This one comes to mind: Damjanovic, K., et al. (2020). "Assessment of bacterial community composition within and among Acropora loripes colonies in the wild and in captivity." Coral Reefs 39(5): 1245-1255. 

We also know that we can reduce the diversity of Exaiptasia in the lab by just changing its environmental conditions: Dungan, A. M., et al. (2021). "Short-Term Exposure to Sterile Seawater Reduces Bacterial Community Diversity in the Sea Anemone, Exaiptasia diaphana." Front Mar Sci 7.

L416: I think you'll find, this really isn't all that surprising. Cnidarians eat a lot of diverse things in the wild, including bacteria (lots of papers here you can cite). When we bring them into the lab we restrict their diet. You can't go back in time to change your experiment, but I would suggest mentioning ways that you could track where these changes (i.e., reduced diversity come from), such as by taking experimental seawater samples, sequencing the Artemia you're feeding your animals, and using approaches that look at the viable microbiota instead of traditional PCR which captures dead bacteria as well as eDNA. 

L419: Yep, you're on the right track. Please add some citations here. 

L421: "Bacteria that are persisting during laboratory maintenance most likely represent bacteria that are functionally associated with N. vectensis and might have co-evolved with its host." I think this is a really interesting idea and would like to see the discussion expanded as such. You could look for "core microbiome members" with the data set you have. 

L430: You mention the Motzfeld et al., 2016 paper several times but fail to compare your data to theirs. What is similar? This goes to my comment above about looking for core microbiome members. You could be sitting on a gold mine. 

L457: You really don't have the dataset to call a section "Microbial plasticity is promoting animal adaptations." You didn't measure any physiology and it would appear that you actually stressed out some animals by shifting them to 20C. 

Overall comments: 

I think there could be an interesting story here, but there are some major issues that need to be addressed first: 

1. The data analysis needs a refresh. This paper is ALL about the N. vectensis microbiota but it was analysed with a tool from 2016-17. So much has changed (for the better) in the last five years. I recognize this is a time-consuming process, but this is the only way that the MS can achieve as high of an impact as PLOS Biology. I would also step away from the OTU approach and look at ASVs (or at least bin OTUs to a greater identity). No stats should be taken from QIIME2 as it does not allow the user to add in random effects. 

2. Genotype becomes a factor in the second experiment; however, 

a. There is no description of the way that the authors determined that animals collected from a same (or different site) were unique genotypes, and

b. There were only two genotypes per site. The authors show that geographic site had a significant impact on microbiota composition (see Fig. 2B). The only "genotypic" differences were between animals from different sites (i.e., Fig. 5F - this is hardly a genotype effect. 

c. Statistically speaking, genotype effects cannot be addressed without considering collection site. 

3. Some experimental decisions need an explanation.

a. The decision to transfer collected animals from their environmental conditions (ranging -2.2 to 22.4 C according to Table S1) and transfer them to 20C.

b. Using two different extraction kits and not including (or presenting) experimental controls in the sequencing run.

4. Discussion needs to be flushed out and address the following topics in more detail:

a. Core microbiome. Which microbiota are essential? How consistent is this? What could they be doing?

b. More of a literature review required on the changes in cnidarian microbial community structure when moved from field to laboratory conditions. There are several papers available. 

Reviewer #2: This manuscript a) describes natural variation across biogeography in the microbial associates of an anemone b) with regard additionally to season and c) clone identity and d) thermal environment (experimental in the laboratory). The analyses are 'broad brush' in that they are based on 16S rRNA sequence, which is a very broad classification tool, and indicate changes in relative abundance, rather than actual abundance. In general the depth of analyses and experiments were strong and patterns were, to me, clear. There is natural latitudinal variation, clone variation, and thermal impacts. 

What is less clear is the broader significance of the findings of this paper. Neither the abstract, introduction nor the discussion gave a compelling case, to me, of why this manuscript reaches beyond a more specialist audience. The microbe changes are not dissected functionally - they are groups of bacteria that change, but how these bacteria live and their metabolism and potential interaction with hosts is not dissected - they remain relatively anonymous taxonomic bins without supporting genomics that might highlighted their biological significance. Functional significance is mentioned only in the discussion in reference to a previously (2021) published preprint. That preprint paper I can see the broader significance of, as it demonstrates a phenotypic functional consequence of the plasticity observed (very interesting); I found this paper more descriptive, essentially reporting clone, thermal and clone x thermal effects on microbiota which are clear, but are to me relatively expected and well executed but routine findings (as is the simplification of microbiota in lab culture, which is commonly seen).

Reviewer #3: The manuscript attempts to address a fundamental question regarding the determinants of microbes associated with animals. This is an essential question because microbes can significantly affect animal physiology, including disease. The research conducts excellent experiments merging natural collections and lab-rearing experiments in Nematostella to tease apart genotype and environmental contributions to the microbiome. In general, I think this work sets the stage for crucial functional analysis of the role of microbes in Nematostella physiology. The experimental design is beautiful, and I hope the authors take the next step and try to discover the roles of particular microbes. 

I have three major questions about the work:

First, it is unclear which animals from which collections were used in each analysis. It may be eluded to or obvious to the writer, but I was counting dots on the plots to determine which animals were included in each figure. This is an essential aspect of the experimental design to clarify. I suggest explicitly adding this information to the figures and results text. For example, if all wild animals were used in the analysis in Fig. 2, genotype may co-vary with collection time. These co-variates may also be addressed through statistical analysis. However, subsetting the data for each comparison is also fair. 

Second, after reading the paper, I was excited to read the end of the discussion (lines 457-471) and the article by Baldassarre 2021 (citation needs updates). I found this work fascinating and directly related to the current study. However, I think the Baldassarre paper needs to be introduced in the introduction and the knowledge gaps highlighted. Then, the results of the present study should be placed in the context of the Baldassarre paper. This would help the non-expert understand how these two studies relate, overlap, and how the field is progressing. 

Third, the data and results are precise. However, the manuscript might benefit from discussing the importance or functional consequences of these shifts in microbes. Are there clear hypotheses generated by these data that may be testable in future studies? What are the major outstanding questions? This will help the reader see the data in a broader context and may inspire the field. 

Minor points:

Lines 32-35: Not really true that five populations of Nematostella were isolated three times during the year. It was only a subset of populations. Minor point, but you may consider clarifying. 

There are two figures 1 and 2. I think they should be 4 and 5. 

Figure 2: It is impossible to tell the differences between many of the colors in this plot. Also, the key is not readable.

---

## [Editor Report · Decision Letter 2]

19 Dec 2022

Dear Dr. Fraune,

Thank you for your patience while we considered your revised manuscript "Genotype-environment interactions determine microbiota plasticity in Nematostella vectensis" for publication as a Research Article at PLOS Biology. This revised version of your manuscript has been evaluated by the PLOS Biology editors and the Academic Editor.

Based on our Academic Editor's assessment of your revision, we are likely to accept this manuscript for publication, provided you satisfactorily address the following data and other policy-related requests.

1. TITLE:

For our broader readership, please could you change your title to "Genotype-environment interactions determine microbiota plasticity in the sea anemone Nematostella vectensis"?

2. DATA POLICY:

Regardless of the method selected, please ensure that you provide the individual numerical values that underlie the summary data displayed in the following figure panels as they are essential for readers to assess your analysis and to reproduce it. We are aware that you have already provided this for some figures, but we will require it for: Figures 1ABCD, 2BCDE, 3ABC, 4ABCD, 5BCDE, 6ABCD, and Supplementary Figures S1, S2, S3AB, S4, S5, S6AB.

**Please also ensure that figure legends in your manuscript include information on where the underlying data can be found, and ensure your supplemental data file/s has a legend.**

We expect to receive your revised manuscript within three weeks.

*Published Peer Review History*

*Press*

Sincerely,

Paula

---

Senior Editor,

pjaureguionieva@plos.org,

PLOS Biology

---

## [Editor Report · Decision Letter 3]

5 Jan 2023

Dear Dr. Fraune,

Thank you for the submission of your revised Research Article "Genotype-environment interactions determine microbiota plasticity in the sea anemone Nematostella vectensis" for publication in PLOS Biology. On behalf of my colleagues and the Academic Editor, Emiley Eloe-Fadrosh, I am pleased to say that we can in principle accept your manuscript for publication, provided you address any remaining formatting and reporting issues. These will be detailed in an email you should receive within 2-3 business days from our colleagues in the journal operations team; no action is required from you until then. Please note that we will not be able to formally accept your manuscript and schedule it for publication until you have completed any requested changes.

PRESS

Sincerely,

Paula

---

Senior Editor

PLOS Biology
